# Guided Discrete Diffusion for Electronic Health Record Generation

**Jun Han\***
*jun_han@optum.com*
*Optum AI, UHG*

**Zixiang Chen\*, Yongqian Li, Yiwen Kou**
*chenzx19@cs.ucla.edu*
*Department of Computer Science, UCLA*

**Eran Halperin, Robert E. Tillman**[†]
*rob.tillman@optum.com*
*Optum AI, UHG*

**Quanquan Gu**[†]
*qgu@cs.ucla.edu*
*Department of Computer Science, UCLA*

**\* Equal contribution** [†] **Project Co-Lead**

**Reviewed on OpenReview:** `https://openreview.net/forum?id=N2rWhTgits`

## Abstract

Electronic health records (EHRs) are a pivotal data source that enables numerous applications in computational medicine, e.g., disease progression prediction, clinical trial design, and health economics and outcomes research. Despite wide usability, their sensitive nature raises privacy and confidentiality concerns, which limit potential use cases. To tackle these challenges, we explore the use of generative models to synthesize artificial, yet realistic EHRs. While diffusion-based methods have recently demonstrated state-of-the-art performance in generating other data modalities and overcome the training instability and mode collapse issues that plague previous GAN-based approaches, their applications in EHR generation remain underexplored. The discrete nature of tabular medical code data in EHRs poses challenges for high-quality data generation, especially for continuous diffusion models. To this end, we introduce a practical tabular EHR generation method, `EHR-D3PM`, which enables both unconditional and conditional generation using the discrete diffusion model. Our experiments demonstrate that `EHR-D3PM` significantly outperforms existing generative baselines on comprehensive fidelity and utility metrics while maintaining less attribute and membership vulnerability risks. Furthermore, we show `EHR-D3PM` is effective as a data augmentation method and enhances performance on downstream tasks when combined with real data.

## 1 Introduction

Electronic health records (EHRs) are a rich and comprehensive data source, enabling numerous applications in computational medicine including the development of models for disease progression prediction and clinical event medical models (Li et al., 2020; Rajkomar et al., 2018), clinical trial design (Bartlett et al., 2019) and health economics and outcome research (Padula et al., 2022). In particular, many existing disease prediction models primarily utilize tabular formats, often transforming longitudinal EHR data into binary or categorical forms, rather than employing time-series forecasting methods (Lee et al., 2022; Huang et al., 2021; Rao et al.,

2023; Debal & Sitote, 2022). However, the sensitive nature of EHRs, which includes confidential medical data, poses challenges for their broad use due to privacy concerns and patient confidentiality requirements (Hodge Jr et al., 1999). In addition to these concerns, data scarcity also restricts their potential use in applications for rare medical conditions. To address these challenges, we consider using generative models to synthesize artificial, but realistic EHRs, which has recently emerged as a crucial research area for advancing applications of machine learning to healthcare and other industries with privacy and data scarcity challenges.

The primary goal of synthetic EHR generation is to generate data that is (i) indistinguishable from real data to an expert, but (ii) not attributable to any actual patients. Recent advancements in deep generative models, including Variational Autoencoders (VAE) (Vincent et al., 2008) and Generative Adversarial Networks (GAN) (Goodfellow et al., 2014), have demonstrated significant promise in generating realistic synthetic EHR data (Biswal et al., 2021; Choi et al., 2017). In particular, GAN-based EHR generation has emerged as the most predominant and popular approach (Choi et al., 2017; Zhang et al., 2020; Torfi & Fox, 2020), and achieved state-of-the-art performance in terms of quality and privacy preservation. However, the unstable training process of GAN-based methods can lead to mode collapse, raising concerns about their widespread application.

Recently, diffusion-based generative models, initially introduced by Sohl-Dickstein et al. (2015), have demonstrated impressive capabilities in generating high-quality samples in various domains, including images (Ho et al., 2020; Song & Ermon, 2020), audio (Chen et al., 2021; Kong et al., 2021), and text (Hoogeboom et al., 2021b; Austin et al., 2021; Chen et al., 2023). A diffusion model consists of a forward process, which gradually transforms training data into pure noise, and a reverse sampling process that reconstructs data from noise using a learned network. Compared to GANs, their training is more stable as it only involves maximizing the log-likelihood of a single neural network.

Due to the superior performance of diffusion models, recent methods have explored their application in generating categorical EHR data (Yuan et al., 2023; Ceritli et al., 2023). While these approaches demonstrate promising performance, their improvement over previous GAN-based methods is varied. Particularly, they struggle to generate EHR records with rare medical conditions at rates consistent with the occurrence of such conditions in real-world data. Furthermore, existing approaches offer limited support for conditional generation, which is crucial for many downstream tasks such as disease classification.

In this paper, we propose a practical EHR generation method that utilizes discrete diffusion (Sohl-Dickstein et al., 2015; Hoogeboom et al., 2021b; Austin et al., 2021; Chen et al., 2023), a type of diffusion process tailored for discrete data sampling, as well as a flexible conditional sampling method that does not require additional model training. Our contributions are summarized as follows:

- We introduce a Discrete Denoising Diffusion model specifically tailored for generation of tabular medical codes in EHRs, dubbed `EHR-D3PM`. Our method incorporates an architecture that effectively captures feature correlations, enhancing the generation process and achieving state-of-the-art performance. Notably, `EHR-D3PM` excels in generating instances of rare conditions, an aspect where existing methods often face challenges.
- We further extend `EHR-D3PM` to conditional generation, specifically tailored for generating EHR samples related to particular medical conditions. Given the unique requirements of this task and the discrete nature of EHR data, we have custom-designed the energy function and applied energy-guided Langevin dynamics at the latent layer of the predictor network to achieve this goal.
- We investigate the effectiveness of `EHR-D3PM` as a data augmentation method in downstream tasks. We show that synthetic EHR data generated by `EHR-D3PM` yields comparable performance to that of real data in terms of AUPRC and AUROC when used to train predictive models and when combined with the real data, `EHR-D3PM` can enhance the performance of predictive models.

**Notation.** We use the symbol $\mathbf{q}$ to denote the real distribution in a diffusion process, while $\mathbf{p}_{\boldsymbol{\theta}}$ represents the distribution parameterized by the NN during sampling. With its success probability inside the parentheses, the Bernoulli distribution is denoted by Bernoulli($\cdot$). We further use Cat($\mathbf{p}$) to denote a categorical distribution over a one-hot row vector with probabilities given by the row vector $\mathbf{p}$.

## 2 Related Work

**EHR Synthesis.** Various methods have been developed for generating synthetic EHR data. Buczak et al. (2010) proposed an early data-driven approach for creating synthetic EHRs, but their approach offers limited flexibility and has privacy concerns. Recently, GANs have become prominent in EHR generation, including medGAN Choi et al. (2017), medBGAN (Baowaly et al., 2018), EHRWGAN (Zhang et al., 2019), and CorGAN Torfi & Fox (2020). GAN-based methods offer significant improvement in the quality of synthetic EHRs, but often face issues related to training instability and mode collapse (Thanh-Tung et al., 2018), restricting their wide use and the diversity of generated data. To address this, other methods, including variational auto-encoders (Biswal et al., 2021) and language models (Wang & Sun, 2022), have been explored. Very recently, MedDiff (He et al., 2023b) considered using diffusion models and proposed sampling techniques for high-quality EHR generation. Ceritli et al. (2023); Yuan et al. (2023) further extended the diffusion model to mixed-type EHRs. In this paper, we focus on developing a guided discrete diffusion model tailored specifically for generating tabular medical codes in Electronic Health Records (EHRs), which has wide-ranging applications in healthcare (Debal & Sitote, 2022; Lee et al., 2022). Our model aims to improve the generation of ICD codes for rare conditions, which has been a challenge for previous methods.

**Discrete Diffusion Models.** Discrete diffusion models was pioneered by Sohl-Dickstein et al. (2015), which explored diffusion processes in binary variables. The approach was further developed by Hoogeboom et al. (2021b), who incorporated categorical random variables using transition matrices with uniform probabilities. Song et al. (2021) introduced a similar extension in their supplementary content, though they did not experiment with this model type. Subsequently, Austin et al. (2021) introduced a generalized framework named Discrete Denoising Diffusion Probabilistic Models (D3PMs) for categorical random variables, effectively combining discrete diffusion models with Masked Language Models (MLMs). Recent advancements in this field include the introduction of editing-based operations (Jolicoeur-Martineau et al., 2021; Reid et al., 2022), auto-regressive diffusion models (Hoogeboom et al., 2021a; Ye et al., 2023), a continuous-time structure (Campbell et al., 2022), strides in generation acceleration (Chen et al., 2023), and the application of neural network analogs for learning purposes (Sun et al., 2023). Discrete diffusion and flow models (Lou et al., 2024; Campbell et al., 2024; Gat et al., 2024) are recently proposed for image, text and protein data (non-tabular). Diffusion models (Kim et al., 2023; Lee et al., 2023; Kotelnikov et al., 2023; Zhang et al., 2024) are recently developed for tabular data generation but not specifically designed for sparse and high dimensional EHR generation. Diffusion and flow-based gradient-boosted trees (Jolicoeur-Martineau et al., 2024) are proposed for tabular data generation but turn out to be extremely slow in high dimension or on large datasets (Cresswell & Kim, 2024). In this paper, we are based on D3PMs with multinomial distribution for EHR generation.

## 3 Background

In this section, we provide background on diffusion models.

**Diffusion Model.** Given $\mathbf{x}_0$ drawn from a target data distribution following $q_{\text{data}}(\cdot)$, the forward process is a Markov process that maps the clean data $\mathbf{x}_0$ to a noisy sample from a prior distribution $q_{\text{noise}}(\cdot)$. The process $\mathbf{x}_0 \to \mathbf{x}_T$ is composed of the conditional distributions $q(\mathbf{x}_t|\mathbf{x}_{t-1}, \mathbf{x}_0)$ where

$$q(\mathbf{x}_{1:T}|\boldsymbol{x}_0) = \prod_{t=1}^{T} q(\mathbf{x}_t|\mathbf{x}_{t-1}, \mathbf{x}_0). \tag{1}$$

By Bayes rule, (1) induces a reverse process $\mathbf{x}_T \to \mathbf{x}_0$ that can convert samples from the prior $q_{\text{noise}}$ into samples from the target distribution $q_{\text{data}}$,

$$q(\mathbf{x}_{t-1}|\mathbf{x}_t, \mathbf{x}_0) = \frac{q(\mathbf{x}_t|\mathbf{x}_{t-1}, \mathbf{x}_0)q(\mathbf{x}_{t-1}|\mathbf{x}_0)}{q(\mathbf{x}_t|\mathbf{x}_0)}. \tag{2}$$

After training a diffusion model, the reverse process can be used for synthetic data generation by sampling from the noise distribution $q_{\text{noise}}$ and repeatedly applying a learnt predictor (neural network) $p_{\boldsymbol{\theta}}(\cdot|\mathbf{x}_t)$ parameterized

by $\boldsymbol{\theta}$:

$$p_{\boldsymbol{\theta}}(\mathbf{x}_T) = q_{\text{noise}}(\mathbf{x}_T), \quad p_{\boldsymbol{\theta}}(\mathbf{x}_{t-1}|\mathbf{x}_t) = \int_{\hat{\mathbf{x}}_0} q(\mathbf{x}_{t-1}|\mathbf{x}_t, \hat{\mathbf{x}}_0) p_{\boldsymbol{\theta}}(\hat{\mathbf{x}}_0|\mathbf{x}_t) d\hat{\mathbf{x}}_0. \tag{3}$$

**Training Objective.** The neural network $p_{\boldsymbol{\theta}}(\cdot|\mathbf{x}_t)$ in (3) that predicts $\hat{\mathbf{x}}_0$ is trained by maximizing the evidence lower bound (ELBO) (Sohl-Dickstein et al., 2015),

$$\log p_{\theta}(\mathbf{x}_0) \geq \mathbb{E}_{q(\mathbf{x}_{1:T}|\mathbf{x}_0)}\left[\log \frac{p(\mathbf{x}_{0:T})}{q(\mathbf{x}_{1:T}|\mathbf{x}_0)}\right]$$

$$= \mathbb{E}_{q(\mathbf{x}_1|\mathbf{x}_0)}[\log p_{\boldsymbol{\theta}}(\mathbf{x}_0|\mathbf{x}_1)] - \sum_{t=2}^{T}\mathbb{E}_{q(\mathbf{x}_t|\mathbf{x}_0)}[\text{KL}(q(\mathbf{x}_{t-1}|\mathbf{x}_t,\mathbf{x}_0)\|p_{\boldsymbol{\theta}}(\mathbf{x}_{t-1}|\mathbf{x}_t))]$$

$$- \mathbb{E}_{q(\mathbf{x}_T|\mathbf{x}_0)}\text{KL}(q(\mathbf{x}_T|\mathbf{x}_0)\|p_{\boldsymbol{\theta}}(\mathbf{x}_T)),$$

Here KL denotes Kullback-Liebler divergence and the last term $\mathbb{E}_{q(\mathbf{x}_T|\mathbf{x}_0)}\text{KL}(q(\mathbf{x}_T|\mathbf{x}_0)\|q_{\text{noise}}(\mathbf{x}_T))$ equals or approximately equals zero if the diffusion process $q$ is properly designed.

Different choices of diffusion process (1) and (2) will result in different sampling methods (3). There are two popular approaches to constructing a diffusion generative model, depending on the nature of the process.

**Gaussian Diffusion Process.** The Gaussian diffusion process assumes a Gaussian noise distribution $\mathbf{q}_{\text{noise}}$. In particular, the prior is chosen to be $q_{\text{noise}} = \mathcal{N}(0, \mathbf{I})$, and the forward process is characterized by

$$q(\mathbf{x}_t|\mathbf{x}_{t-1}, \mathbf{x}_0) = \mathcal{N}(\mathbf{x}_t; \sqrt{1-\beta_t}\mathbf{x}_{t-1}, \beta_t\mathbf{I}),$$

where $\beta_t$ is the variance schedule determined by a pre-specified corruption schedule. The Gaussian diffusion process has achieved great success in continuous-valued applications like image generation (Ho et al., 2020; Song et al., 2021). Recently, it has been applied to tabular EHR data generation (He et al., 2023b; Yuan et al., 2023).

**Discrete Diffusion Process.** Discrete Denoising Diffusion Probabilistic Models (D3PMs) is designed to generate categorical data from a vocabulary $\{1, \ldots, K\}$, represented as a one-hot vector $\mathbf{x} \in \{0, 1\}^K$. The noise follows a categorical distribution $\mathbf{q}_{\text{noise}}$. The Multinomial distribution (Hoogeboom et al., 2021b) is among the most effective noise distributions. In particular, $\mathbf{q}_{\text{noise}}$ is chosen to be a uniform distribution over the one-hot basis of the vocabulary $\{\mathbf{e}_1, \ldots, \mathbf{e}_K\}$, and the forward process is characterized by

$$q(\mathbf{x}_t|\mathbf{x}_{t-1}, \mathbf{x}_0) = \text{Cat}\big(\mathbf{x}_t; \beta_t\mathbf{x}_{t-1} + (1-\beta_t)\mathbf{q}_{\text{noise}}\big),$$

where Cat is the categorical distribution and $\beta_t$ is the variance schedule determined by a pre-specified corruption schedule. Due to its discrete nature, D3PM is widely used to generate categorical data like text (Hoogeboom et al., 2021b; Austin et al., 2021) and categorical tabular data Kotelnikov et al. (2023); Ceritli et al. (2023). This paper uses a D3PM with a multinomial noise distribution to generate tabular medical codes in EHRs.

# 4 Method

In this section, we formalize the problem of tabular EHR data generation and provide the technical details of our method.

## 4.1 Problem Formulation

We consider medical coding data in EHRs, such as diagnose codes (ICD), procedure codes (CPT) and medication codes (GEN), which are standardized medical codes (Slee, 1978). In a wide application of medical domains, it is common to convert continuous variables into discrete ones to enhance the performance of prediction models (Rasmy et al., 2021; Hill et al., 2023). For example, Hill et al. (2023) converts continuous lab codes to discrete tokens based on deciles of each LOINC code; prediction models built on learnt representation

of tokenized discrete data significantly outperform ML models which are directly trained on the mixed-type tabular data. Therefore, our paper focus on the generation of discrete medical codes, e.g., ICD, CPT, GEN and LOINC codes.

For a given set $\Omega$ of medical codes of interest, we encode the set as $N := |\Omega|$ categories $\{1, 2, \ldots, N\}$. A sample patient EHR $\mathbf{x}$ is then encoded as a sequence of $N$ tokens $\mathbf{x} = [\mathbf{x}^{(1)}, \ldots, \mathbf{x}^{(N)}]$, where each token $\mathbf{x}^{(i)} \in \{0,1\}^2$ is a one-hot function. $\mathbf{x}^{(i)}$ represents the occurrence of the $i$-th medical code in the patient EHR. Although the total number of medical codes is large, the number of medical codes each code is correlated with should be relatively small. Therefore, the joint distribution on $\Omega$ can be approximately factorized and the intrinsic dimension of the data is low. We assume a sufficiently large set of patient EHRs is available to train a transformer-based diffusion model to generate synthetic EHRs sequences $\mathbf{x}'$.

## 4.2 Unconditional Generation

In Section 3, we introduced multinomial diffusion with a single token, $\mathbf{x} \in \mathbb{R}^K$. In the context of categorical EHRs, we aim to generate a sequence of $N$ tokens with $K = 2$, denoted by $\mathbf{x} = [\mathbf{x}^{(1)}, \ldots, \mathbf{x}^{(N)}]$. Therefore, we need to extend the terminology from Section 3. We define the sequence of tokens at the $t$-th time step as $\mathbf{x}_t = [\mathbf{x}_t^{(1)}, \ldots, \mathbf{x}_t^{(N)}]$, where $\mathbf{x}_t^{(i)}$ represents the $i$-th token at diffusion step $t$. Multinomial noise $\mathbf{q}_{\text{noise}}$ is added to each token in the sequence independently during the diffusion process,

$$q(\mathbf{x}_t | \mathbf{x}_{t-1}, \mathbf{x}_0) = \prod_{i=1}^N \text{Cat}\big(\mathbf{x}_t^{(i)}; \beta_t \mathbf{x}_{t-1}^{(i-1)} + (1 - \beta_t) \mathbf{q}_{\text{noise}}\big).$$

The reverse sampling procedure uses the predictor $p_{\boldsymbol{\theta}}(\cdot | \mathbf{x}_t)$ with the following neural network architecture with $\mathbf{z}_{0,t} = \mathbf{x}_t = [\mathbf{x}_t^{(1)}, \ldots, \mathbf{x}_t^{(N)}]$,

$$\mathbf{z}'_{l,t} = \mathbf{z}_{l-1,t} + \mathbf{E}_{\text{pos}} + \mathbf{E}_{\text{time}}, \qquad \mathbf{z}_{l,t} = \text{LinMSA}(\text{LN}(\mathbf{z}'_{l-1,t})) + \mathbf{z}'_{l-1,t},$$
$$\mathbf{z}_{L,t} = \text{MLP}(\text{LN}(\mathbf{z}_{L-1,t})), \qquad \text{Output} = \text{Softmax}(\mathbf{z}_{L,t}), \tag{4}$$

where $\mathbf{E}_{\text{pos}}, \mathbf{E}_{\text{time}} \in \mathbb{R}^{2 \times D}$ represents the position embedding and time embedding respectively, the variable $l$ indexes the layers belonging to the set $\{1, \ldots, L\}$. LinMSA refers to the efficient multi-head self-attention block proposed by Wang et al. (2020), which has linear complexity with respect to the input dimension. LN is an abbreviation for layer normalization. For each dimension of $\hat{\mathbf{x}}_0$, we apply a multilayer perceptron layer to obtain the logit, abbreviated as ParallelMLP. The softmax function transforms the last-layer latent variable $\mathbf{z}_L^i$ into the conditional probability $p_{\boldsymbol{\theta}}(\cdot | \mathbf{x}_t)$, serving as the final softmax layer. The details of our denoise model are provided in Figure 4 in Appendix A.2.

## 4.3 Conditional Generation with Classifier Guidance

The goal of conditional generation is to generate $p_{\boldsymbol{\theta}}(\mathbf{x} | \mathbf{c})$ close to $q_{\text{data}}(\mathbf{x} | \mathbf{c})$, where $\mathbf{c}$ denotes a context, such as the presence of a single or group of medical codes in a patient's Electronic Health Record (EHR). Since $\mathbf{c}$ is not available during the training process, we cannot train the generator $p_{\boldsymbol{\theta}}(\mathbf{x} | \mathbf{c})$ directly. However, we assume access to a classifier $p(\mathbf{c} | \mathbf{x})$ that approximates the conditional distribution $q_{\text{data}}(\mathbf{c} | \mathbf{x})$. Given an unconditional EHR generator $p_{\boldsymbol{\theta}}(\mathbf{x})$ and classifier $p(\mathbf{c} | \mathbf{x})$, we can propose a training-free conditional generator as follows:

$$p_{\boldsymbol{\theta}}(\mathbf{x} | \mathbf{c}) \propto p_{\boldsymbol{\theta}}(\mathbf{x}) \cdot p(\mathbf{c} | \mathbf{x}). \tag{5}$$

Since $q_{\text{data}}(\mathbf{x} | \mathbf{c}) \propto q_{\text{data}}(\mathbf{x}) \cdot q_{\text{data}}(\mathbf{c} | \mathbf{x})$, we can expect $p_{\boldsymbol{\theta}}(\mathbf{x} | \mathbf{c})$ in (5) is close to $q_{\text{data}}(\mathbf{x} | \mathbf{c})$ provided the unconditional generator $p_{\boldsymbol{\theta}}(\mathbf{x})$ is close to $q_{\text{data}}(\mathbf{x})$ and the classifier $p(\mathbf{c} | \mathbf{x})$ is close to $q_{\text{data}}(\mathbf{c} | \mathbf{x})$.

To sample from equation (5), the most popular approach is applying Langevin sampling on the unnormalized joint density while injecting Gaussian noise (Ho & Salimans, 2021). However, there is a challenge in multinomial diffusion as the state $\mathbf{x}$ lies in a discrete space. In particular, the multinomial diffusion procedure is as follows:

$$p_{\boldsymbol{\theta}}(\mathbf{x}_{t-1} | \mathbf{x}_t, \mathbf{c}) = \sum_{\hat{\mathbf{x}}_0} q(\mathbf{x}_{t-1} | \hat{\mathbf{x}}_0, \mathbf{x}_t) p_{\boldsymbol{\theta}}(\hat{\mathbf{x}}_0 | \mathbf{x}_t, \mathbf{c}),$$

where $\hat{\mathbf{x}}_0$ is the latent variable that predicts $\mathbf{x}_0$. It is not possible to directly apply Langevin dynamics in the space of $\hat{\mathbf{x}}_0$. However, the last-layer latent variable $\mathbf{z}_{L,t}$ in Equation (4) (before the softmax layer) lies in a continuous space. We have the following:

$$p_{\boldsymbol{\theta}}(\hat{\mathbf{x}}_0 \mid \mathbf{x}_t, \mathbf{c}) = \int p_{\boldsymbol{\theta}}(\hat{\mathbf{x}}_0 \mid \mathbf{z}_{L,t})\, p_{\boldsymbol{\theta}}(\mathbf{z}_{L,t} \mid \mathbf{x}_t, \mathbf{c})\, d\mathbf{z}_{L,t}.$$

Therefore, we can utilize the plug-and-play method, initially employed in text generation (Dathathri et al., 2019) and recently applied to protein design (Gruver et al., 2023), with the latent space $\mathbf{z}_L$. In particular, we introduce a modified latent variable $\mathbf{y}^{(k)}$ for $\mathbf{z}_{L,t}$, which is initialized as $\mathbf{y}^{(0)} \leftarrow \mathbf{z}_{L,t}$. This modification allows us to iteratively update the latent variable using Langevin dynamics, guiding it towards a desired target while maintaining the learned structure of the diffusion model. The iterative update is applied as follows:

$$\mathbf{y}^{(k+1)} \leftarrow \mathbf{y}^{(k)} - \eta \nabla_{\mathbf{y}^{(k)}} [\mathcal{D}_{\mathrm{KL}}(\mathbf{y}^{(k)}) - V_{\boldsymbol{\theta}}(\mathbf{y}^{(k)})] + \sqrt{2\eta\tau}\boldsymbol{\epsilon},$$

where the energy function $V_{\boldsymbol{\theta}}(\mathbf{y}^{(k)}) = \log(p(\mathbf{c}|\mathbf{y}^{(k)})) = \log\left(\sum_{\hat{\mathbf{x}}_0} p_{\boldsymbol{\theta}}(\hat{\mathbf{x}}_0|\mathbf{y}^{(k)})p(\mathbf{c}|\hat{\mathbf{x}}_0)\right)$ and $\mathcal{D}_{\mathrm{KL}}(\mathbf{y}^{(k)}) = \lambda \mathrm{KL}(p_{\theta}(\hat{\mathbf{x}}_0|\mathbf{y}^{(k)})||p_{\theta}(\hat{\mathbf{x}}_0|\mathbf{y}^{(0)}))$ is the Kullback–Leibler (KL) divergence for regularization of the guided Markov transition. The gradient of the energy term $\nabla_{\mathbf{y}^{(k)}} V_{\boldsymbol{\theta}}$ drives the hidden state $\mathbf{y}^{(k)}$ towards high probability of $p(\mathbf{c}|\mathbf{y}^{(k)})$, ensuring that the generated samples align with the desired target. The gradient of the regularization term $\nabla_{\mathbf{y}^{(k)}} \mathcal{D}_{\mathrm{KL}}$ ensures that the guided transition distribution still maximizes the likelihood of the diffusion model, preserving the learned structure and preventing excessive deviation from the original model. For a more detailed discussion, see Appendix C.

## 5 Experiments

In this section, we apply our method to three EHR datasets, including the widely used public MIMIC-III dataset and two larger private datasets from a large health institute[1]. We compare our method with state-of-the-art EHR generative models in terms of fidelity, utility and privacy.

### 5.1 Experiment Setup

**Datasets. Public Datasets** MIMIC-III (Johnson et al., 2016) includes deidentified patient EHRs from hospital stays. For each patient's EHR, we extract the diagnosis and procedure ICD-9 codes and truncate the codes to the first three digits. This dataset includes a patient population of size 46,520. **Private Datasets** We consider two industrial EHR datasets. $\mathcal{D}_1$ includes a patient population of size 1,670,347 and has sparse binary features. $\mathcal{D}_2$ includes a population of size 1,859,536 and has relatively denser binary features from a different corpus. $\mathcal{D}_2$ contains diagnose codes (ICD), procedure codes (CPT) and medication codes (GEN).

**Diseases of Interest.** We consider the utility of the synthetic data in six chronic diseases: type-II diabetes, chronic obstructive pulmonary disease (COPD), chronic kidney disease (CKD), asthma, hypertension heart and osteoarthritis.

### 5.2 Baselines

**Med-WGAN**, **EMR-WGAN** and **EHRMGAN** A number of GAN models (Choi et al., 2017; Baowaly et al., 2018; Torfi & Fox, 2020) have been proposed for realistic EHR generation. Med-WGAN and EMR-WGAN have best performance among GAN-based methods. EHRMGAN is one of most recent GAN-based method whose repository is public and incorporates the training of conditional generation. Therefore, we compare our guided generation with the conditional generation of EHRMGAN.

**TabDDPM** and **TabSyn** We compare our model with two recent diffusion models, TabDDPM (Kotelnikov et al., 2023) and TabSyn (Zhang et al., 2024) for tabular data. When training TabSyn in first 100 dimensions of our datasets, TabSyn works well. But when training TabSyn in first 200 dimensions or all dimensions of

---

[1]To comply with the double-blind submission policy, we withhold the name of the institution providing the datasets; should the paper be accepted, we will provide these details.

our datasets, the performance of **TabSyn** degenerates significantly. One reason could be that the feature in EHRs are extremely sparse, which is challenging for learning in the first component of TabSyn. Therefore we leave the comparison with TabSyn on Appendix B.1.

**EHRDiff** Yuan et al. (2023) is the only diffusion model directly designed for synthesizing tabular EHR with an open-source codebase. As the code of other diffusion models (Yuan et al., 2023; Ceritli et al., 2023; He et al., 2023a) for tabular EHRs are not available, we select EHRDiff as a baseline. EHRDiff uses Gaussian distributions in DDPM and apply simple MLP networks in the denoising model.

**ForestDiff** The improved version (Cresswell & Kim, 2024) of ForestDiff (Jolicoeur-Martineau et al., 2024) is trained for comparison, which is still extremely slow in high dimension or large dataset as only CPUs can be used for training and sampling. Therefore, we only provide its comparison on MIMIC dataset on Appendix B.

### 5.3 Evaluation Metrics

**Dimension-wise Prevalence** We compute dimension-wise prevalence by taking the mean of the data in each dimension. Dimension-wise prevalence captures the marginal feature distribution of the data. We compute the Spearman correlation between prevalence in the synthetic data and prevalence in the real data.

**Correlation Matrix Distance** (CMD) measures the difference between the covariance matrix of the synthetic data and the covariance matrix of real data. We first compute the empirical covariance matrices of the synthetic data and real data respectively and take the difference between these two matrices. Then we calculate the Frobenius norm of the difference matrix as distributional distance.

**Maximum Mean Discrepancy** (MMD) is one of most common metrics to measure the difference between two distributions in distributional space. We compute the MMD for a set of synthetic data and a set of real test data. We employ a mixture of kernel-based methods (Li et al., 2017) to estimate MMD to improve its robustness. The detailed formula is given in Eq. (6) on Appendix A.4.

**Medical Concept Abundance Distance** (MCAD) measures the synthetic EHR distribution on record level. It quantifies the discrepancy between the empirical distribution (histogram) of the unique positive code number on synthetic data and the empirical distribution of the unique positive code number on real data.

**Downstream Prediction** (**AUPRC** and **AUROC**) To evaluate the utility of the generated data, we evaluate the accuracy of classifiers trained to predict the diseases of interest mentioned above using synthetic data. We train the classifiers to predict the ICD code that corresponds to the disease of interest using all other available ICD codes as features. We train the classification model using synthetic data and evaluate its performance on real test data. We adopt the state-of-the-art robust classification model for tabular data given in (Ke et al., 2017). The most reliable classification model is one trained on real data; we use this as a benchmark to represent an upper bound for classification accuracy.

**Precision** and **Recall** To evaluate the quality and coverage of synthetic samples by conditional sampling methods, precision and recall metrics induced from Kynkäänniemi et al. (2019) is computed, where we use elementwise $L_1$ distance to measure the distance between two samples.

**Privacy Metrics** Attribute inference risk (AIR) and membership inference risk (MIR) are used to evaluate the vulnerability risks. AIR measures the risk from the adversary ability to infer sensitive attributes of a targeted record when a subset of features are exposed to the attacker. AIR is calculated as the weighted sum of F1 scores of the inferences of other sensitive attributes. The number of exposed attributes for all experiments is defaulted as 256. MIR evaluates the risk that an attacker can infer the real samples used for training the generative model given generated EHR data or the model parameters. For each EHR in this set of real training and evaluation data, we calculate the minimum L2 distance with respect to the synthetic EHR data. The real EHR whose distance is smaller than a preset threshold is predicted as the training EHR. The predicted F1 score is computed to evaluate membership vulnerability risk. The threshold is set as 3 for all experiments. We use codebase of Yan et al. (2022) to compute both AIR and MIR scores.

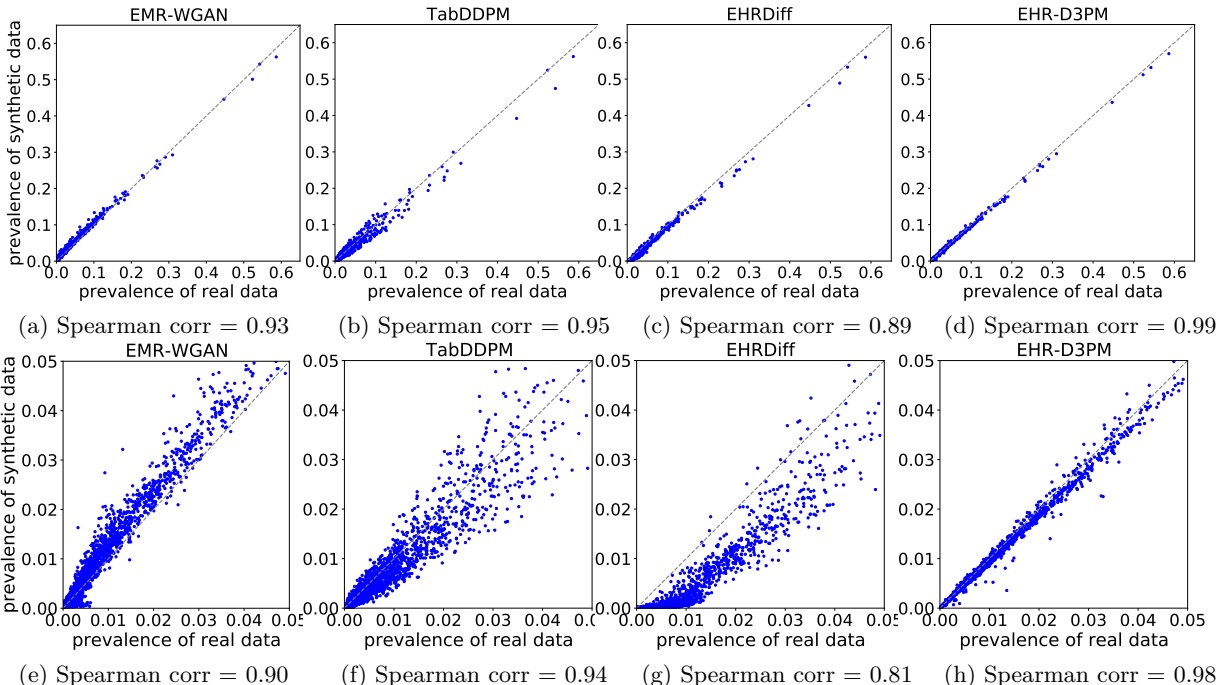

Figure 1: Comparison of prevalence on synthetic data and real data $\mathcal{D}_2$ with ICD, CPT and GEN codes, where the total dimension is 2683. The second row represents the prevalence of the first row in the low data regime. The prevalence is computed on 200K samples. The dashed diagonal lines represent the perfect matching of code prevalence between synthetic data and real EHR data. Pearson correlations are very high for all methods and thus not used as a metric to compare different methods.

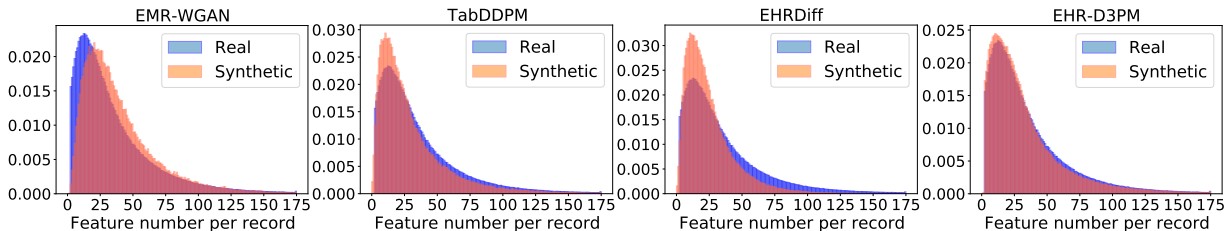

Figure 2: Density comparison of per-record feature number on synthetic data and real data $\mathcal{D}_2$ with ICD, CPT and GEN codes. The number of features per record is the sum of ICD codes present in each sample. The number of bins is 175, and the range of feature number values is (0, 175).

## 5.4 Experiment Results

**Fidelity**  We first evaluate the accuracy of marginal distributions on synthetic data in Figure 1 for dataset $\mathcal{D}_2$, Figure 5 for MIMIC and Figure 7 for $\mathcal{D}_1$ on Appendix B.1. We compare prevalence in synthetic data with prevalence in real data for each dimension. In Figure 1, Figure 5, and Figure 7, we can see that the prevalence for our method EHR-D3PM aligns best with the real data. EHR-D3PM consistently has the highest Spearman correlation. We further observe that EMR-WGAN and EHRDiff fail to provide an unbiased estimation of the distribution in the low prevalence regime, which corresponds to rare conditions. This failure is mild when the dataset has dense features, as shown in Figure 1 of $\mathcal{D}_2$, but is obvious when the dataset has sparse features, as shown in Figure 7 of $\mathcal{D}_1$.

Next we visualize the histogram of feature number per record, which is calculated by summing the medical codes in each sample, on dataset $\mathcal{D}_2$ in Figure 2, MIMIC-III dataset in Figure 6, and dataset $\mathcal{D}_1$ in Figure 8 on Appendix B.1. We compare feature numbers for synthetic data with that of real data. As we see in Figure 6, Figure 8 and Figure 2, EMR-WGAN and EHRDiff demonstrate poor performance in estimating the mode or the tail of the density. When the datasets are large, our method tends to provide a perfect estimation of the feature number for the real data.

Finally we compare our method with baselines in fidelity metrics, CMD, MMD and MCAD in Table 1. The results in CMD metric show that our EHR-D3PM significantly outperforms all baselines, particularly with the larger datasets $\mathcal{D}_1$ and $\mathcal{D}_2$. This indicates our method can learn much better pairwise correlations between different feature dimensions. Table 1 also shows that the distribution of synthetic data learned by our method has the least discrepancy with real data distribution on all three datasets.

Table 1: Fidelity metrics (CMD, MMD and MCAD) on MIMIC and real dataset $\mathcal{D}_2$. These metrics on real dataset $\mathcal{D}_1$ are provided on Appendix due to space limit.

| METRICS | CMD($\downarrow$) | | MMD($\downarrow$) | | MCAD($\downarrow$) | |
|---|---|---|---|---|---|---|
| DATASETS | MIMIC | $\mathcal{D}_2$ | MIMIC | $\mathcal{D}_2$ | MIMIC | $\mathcal{D}_2$ |
| MED-WGAN | $27.540_{\pm 0.628}$ | $28.942_{\pm 0.196}$ | $0.078_{\pm 0.0089}$ | $0.086_{\pm 0.013}$ | $0.1896_{\pm 0.0024}$ | $0.1944_{\pm 0.0017}$ |
| EMR-WGAN | $26.658_{\pm 0.639}$ | $21.438_{\pm 0.146}$ | $0.053_{\pm 0.0054}$ | $0.024_{\pm 0.004}$ | $0.1546_{\pm 0.0017}$ | $0.1572_{\pm 0.0015}$ |
| TABDDPM | $25.236_{\pm 0.684}$ | $16.728_{\pm 0.126}$ | $0.010_{\pm 0.0016}$ | $0.042_{\pm 0.008}$ | $0.1306_{\pm 0.0018}$ | $0.1587_{\pm 0.0018}$ |
| EHRDIFF | $25.447_{\pm 0.485}$ | $18.941_{\pm 0.092}$ | $0.009_{\pm 0.0013}$ | $0.046_{\pm 0.005}$ | $0.1439_{\pm 0.0015}$ | $0.1764_{\pm 0.0016}$ |
| EHR-D3PM | $\mathbf{21.128}_{\pm 0.393}$ | $\mathbf{10.255}_{\pm 0.037}$ | $\mathbf{0.003}_{\pm 0.0004}$ | $\mathbf{0.019}_{\pm 0.002}$ | $\mathbf{0.1013}_{\pm 0.0010}$ | $\mathbf{0.1081}_{\pm 0.0009}$ |

Table 2: Synthetic data utility. Disease prediction from ICD codes on real data $\mathcal{D}_2$. AUROC is reported. We use synthetic data of 160K to train the classifier and 200K real test data to evaluate different methods. 80% of test data are bootstrapped 50 times to compute 95% confidence interval.

| | T2D | ASTHMA | COPD | CKD | HTN-HEART | OSTEOARTHRITIS |
|---|---|---|---|---|---|---|
| REAL DATA | $0.955_{\pm 0.001}$ | $0.853_{\pm 0.002}$ | $0.951_{\pm 0.002}$ | $0.944_{\pm 0.002}$ | $0.926_{\pm 0.003}$ | $0.893_{\pm 0.002}$ |
| MED-WGAN | $0.924_{\pm 0.001}$ | $0.819_{\pm 0.002}$ | $0.853_{\pm 0.003}$ | $0.835_{\pm 0.004}$ | $0.500_{\pm 0.001}$ | $0.820_{\pm 0.003}$ |
| EMR-WGAN | $0.918_{\pm 0.001}$ | $0.747_{\pm 0.002}$ | $0.888_{\pm 0.003}$ | $0.907_{\pm 0.003}$ | $0.844_{\pm 0.005}$ | $0.753_{\pm 0.003}$ |
| TABPPDM | $0.945_{\pm 0.003}$ | $0.846_{\pm 0.003}$ | $0.931_{\pm 0.004}$ | $0.915_{\pm 0.009}$ | $0.837_{\pm 0.008}$ | $0.868_{\pm 0.005}$ |
| EHRDIFF | $0.950_{\pm 0.001}$ | $0.843_{\pm 0.002}$ | $0.936_{\pm 0.002}$ | $0.916_{\pm 0.004}$ | $0.822_{\pm 0.006}$ | $0.875_{\pm 0.002}$ |
| EHR-D3PM | $\mathbf{0.952}_{\pm 0.001}$ | $\mathbf{0.853}_{\pm 0.002}$ | $\mathbf{0.947}_{\pm 0.002}$ | $\mathbf{0.944}_{\pm 0.002}$ | $\mathbf{0.911}_{\pm 0.004}$ | $\mathbf{0.889}_{\pm 0.002}$ |

**Utility** We now apply our method to disease classification downstream tasks. Since MIMIC-III contains a much smaller patient population than the private datasets, which may not provide a valid test data benchmark for disease classification, we focus on the private datasets $\mathcal{D}_1$ and $\mathcal{D}_2$. In Table 5, we observe that the prevalence of most diseases is low in datasets $\mathcal{D}_1$ and $\mathcal{D}_2$. The training and test data set sizes are 160K and 200K. From Table 2 and Table 9 on Appendix B.1, the absolute average increase in AUPRC and AUROC over baselines (EHRDiff and TabDDPM) is 3.22% and 3.12% respectively for dataset $\mathcal{D}_2$. From Table 7 and Table 8 on Appendix B.1, the absolute average increase in AUPRC and AUROC over baselines (EHRDiff and TabDDPM) is 3.90% and 2.57% respectively for dataset $\mathcal{D}_1$. The improvement in downstream prediction for our model benefits from the best correlation between different features our model has learned.

**Privacy** In Table 3, we evaluate privacy metrics AIR and MIR. It shows that our model has relatively low risks on both AIR and MIR. This indicates our method has mild vulnerability to privacy risk compared to existing baselines. Generally, there is a trade-off between privacy and fidelity for generative models. In particular, when a generative model fails to learn any information of the data distribution, AIR and MIR scores are expected to be the lowest. Therefore, incorporating differential privacy to further reduce the privacy risk in discrete diffusion models is an interesting direction, which is largely unexplored.

Table 3: Privacy metrics (AIR and MIR) on MIMIC, dataset $\mathcal{D}_1$ and dataset $\mathcal{D}_2$.

| | ATTRIBUTE INFERENCE RISK($\downarrow$) | | | MEMBERSHIP INFERENCE RISK($\downarrow$) | | |
| --- | --- | --- | --- | --- | --- | --- |
| | MIMIC | $\mathcal{D}_1$ | $\mathcal{D}_2$ | MIMIC | $\mathcal{D}_1$ | $\mathcal{D}_2$ |
| MED-WGAN | $0.019_{\pm 0.0015}$ | $0.071_{\pm 0.0056}$ | $0.080_{\pm 0.0056}$ | $0.440_{\pm 0.0034}$ | $0.339_{\pm 0.0018}$ | $0.398_{\pm 0.0019}$ |
| EMR-WGAN | $0.058_{\pm 0.0027}$ | $0.116_{\pm 0.0084}$ | $0.132_{\pm 0.0089}$ | $0.456_{\pm 0.0035}$ | $0.358_{\pm 0.0017}$ | $0.415_{\pm 0.0020}$ |
| TABDDPM | $0.021_{\pm 0.0013}$ | $0.085_{\pm 0.0091}$ | $0.093_{\pm 0.0062}$ | $0.462_{\pm 0.0037}$ | $0.351_{\pm 0.0018}$ | $0.415_{\pm 0.0020}$ |
| EHRDIFF | $0.022_{\pm 0.0016}$ | $0.077_{\pm 0.0042}$ | $0.089_{\pm 0.0054}$ | $0.445_{\pm 0.0034}$ | $0.353_{\pm 0.0015}$ | $0.421_{\pm 0.0019}$ |
| EHR-D3PM | $0.020_{\pm 0.0014}$ | $0.068_{\pm 0.0046}$ | $0.078_{\pm 0.0048}$ | $0.432_{\pm 0.0034}$ | $0.344_{\pm 0.0016}$ | $0.406_{\pm 0.0018}$ |

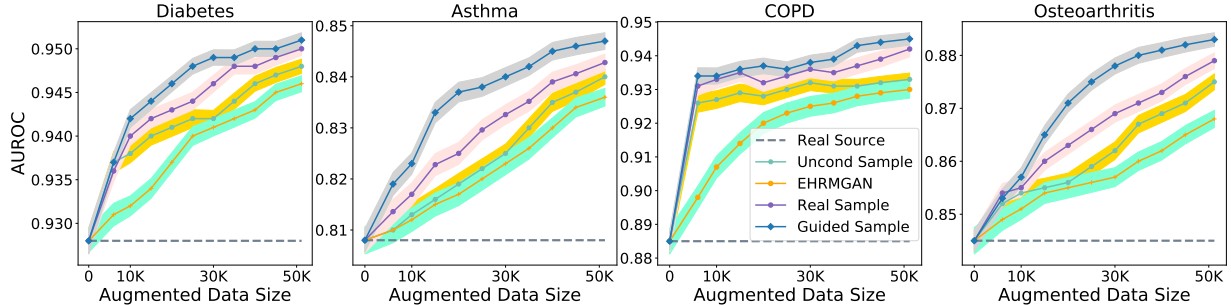

Figure 3: Synthetic data augmentation for disease classifications from ICD codes based on dataset $\mathcal{D}_2$. The size of the real source data for training the LGBM classifier is 5000, as indicated by the dashed purple line. We augment the source training data with synthetic data to train the LGBM classifier. "Uncond Samples" stands for the synthetic data generated by our unconditional sampler. Guided samples are synthetic data generated by our proposed guided sampler for each disease. To minimize noise from evaluation, we adopt 200K real test data to evaluate all experiments and report test AUROC for comparison. 80% of the test data are bootstrapped 50 times to compute 95% CI, which is visualized by the shaded region around each line.

## 5.5 Guided Generation

In this section, we apply our guided sampling method to generate conditional samples for different disease conditions. For each condition, we apply our guided sampler and generate a set of synthetic data. We evaluate the precision and recall of generated samples based on Kynkäänniemi et al. (2019) in Table 4. As we can see, our guided sampler has consistently higher recall in diabetes, COPD, Asthma and Osteoarthritis. Compared with our unconditional method, samples of the conditional baseline EHRMGAN have slightly better recall score but much worse precision score.

Table 4: Precision ($\uparrow$) and Recall ($\uparrow$) for samples of different diseases on $\mathcal{D}_2$. "Uncond" denotes samples generated by our unconditional method. "Guided" stands for samples generated by our guided sampler.

| | DIABETES | | ASTHMA | | COPD | | OSTEOARTHRITIS | |
| --- | --- | --- | --- | --- | --- | --- | --- | --- |
| | PRECISION | RECALL | PRECISION | RECALL | PRECISION | RECALL | PRECISION | RECALL |
| UNCOND | $0.785_{\pm .013}$ | $0.724_{\pm .014}$ | $0.657_{\pm .011}$ | $0.416_{\pm .007}$ | $0.708_{\pm .009}$ | $0.406_{\pm .007}$ | $0.436_{\pm .010}$ | $0.212_{\pm .011}$ |
| EHRMGAN | $0.636_{\pm .015}$ | $0.778_{\pm .016}$ | $0.516_{\pm .014}$ | $0.474_{\pm .012}$ | $0.592_{\pm .009}$ | $0.483_{\pm .010}$ | $0.335_{\pm .012}$ | $0.239_{\pm .013}$ |
| GUIDED | $0.752_{\pm .011}$ | $\mathbf{0.863_{\pm .013}}$ | $0.574_{\pm .010}$ | $\mathbf{0.591_{\pm .008}}$ | $0.653_{\pm .009}$ | $\mathbf{0.617_{\pm .006}}$ | $0.404_{\pm .009}$ | $\mathbf{0.316_{\pm .009}}$ |

In the following, we utilize synthetic samples to augment training data in downstream disease classifiers. The size of the real source data for training classifiers is 5000. We augment the original training data with data from three different groups: real data, synthetic data generated by our unconditional sampling method and our guided sampling method. We report AUROC in Figure 3 and AUPRC in Figure 9 on Appendix B.2 to

evaluate accuracy. We can see that classifiers trained with synthetic data augmentation always improve the vanilla baseline (classifier trained with the original source data). We also observe that the data augmentation by guided sampling consistently outperform the data augmentation by our unconditional sampling and the conditional baseline EHRMGAN. It is interesting to observe that the data augmentation by guided sampling has consistently higher AUROC than real data augmentation. We observe this to be because the synthetic samples generated by guided sampling contain richer information in cases of diseases of low prevalence.

## 6 Conclusion and Future Work

In this paper, we introduced a practical generative model for synthesizing realistic EHRs `EHR-D3PM`. Leveraging the latest advancements in discrete diffusion models, `EHR-D3PM` overcomes the challenges of GAN-based approaches and effectively generates high-quality tabular medical data. Compared with other diffusion-based approaches, `EHR-D3PM` enables high-quality conditional generation. Our experiment demonstrates that `EHR-D3PM` not only achieves state-of-the-art performance in terms of fidelity, utility, and privacy metrics but also significantly improves downstream task performance through data augmentation. Incorporating longitudinal feature into our model is an interesting research direction. Further investigations of the vulnerability of diffusion-based generative models in EHR generation, particularly to attribute and membership inference attacks (Shokri et al., 2017), is a promising future direction as well as providing formal privacy guarantees, e.g., by incorporating differential privacy, which is largely unexplored in diffusion-based models for discrete data.

### Broader Impacts

The primary goal of this work was to develop a state-of-the-art generative model for medical code data in EHRs. The synthetic data generated by this model may be used to train, and augment the training of various downstream models for computational medicine. We believe this work may enable future applications of machine learning to healthcare which lead to positive impacts for patients, e.g., through early prediction of disease progression or matching patients to appropriate clinical trials that advance medicine. While the use of generative models and synthetic data can enhance patient privacy, they do not eliminate the risks of membership inference and other attacks, as we mention and demonstrate in the paper. Our experiments demonstrate that the proposed model achieves state-of-the-art performance in terms of reducing these risks, but proper controls must still be used whenever working with patient data. Furthermore, any downstream models trained using synthetic data must go through the same rigorous review and testing process to prevent disparate impact as models trained on real data.

### Limitations

Our proposed `EHR-D3PM` model focuses on generating synthetic tabular EHRs and achieves state-of-the-art performance, which has a significant impact on various tasks. However, there are certain limitations to our work that should be acknowledged.

Firstly, our model does not currently incorporate longitudinal features, which could potentially enhance the utility of the generated data. Incorporating temporal dependencies and modeling the evolution of patient records over time is an important direction for future research. This extension would enable the generation of more realistic and comprehensive EHR data, capturing the dynamic nature of patient health over extended periods.

Secondly, the vulnerability of diffusion-based generative models in EHR generation to more advanced privacy attacks is a largely unexplored area that requires further research. While our model demonstrates strong privacy preservation properties, it is crucial to investigate its resilience against sophisticated adversarial attacks specifically designed to target EHR data. Thorough analysis and development of robust defense mechanisms are necessary to ensure the long-term security and privacy of the generated synthetic EHRs.

By acknowledging these limitations, we aim to provide a transparent assessment of our work's scope and potential areas for future exploration.

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

## A   Experiment Details.

In the following Table 5, we present a concise summary of various diseases along with their corresponding International Classification of Diseases, Ninth Revision (ICD 9) codes. This table includes common conditions such as Type II Diabetes (T2D), Chronic Kidney Disease (CKD), Chronic Obstructive Pulmonary Disease (COPD), Asthma, Hypertension and Osteoarthritis. Each disease is associated with specific ICD 9 codes that are used for clinical classification and diagnosis purposes. In this paper, we are interested in the diseases listed in Table 5.

Table 5: List of Diseases and Corresponding ICD 9 Codes.

| Disease | ICD 9 Code | MIMIC | $\mathcal{D}_1$ | $\mathcal{D}_2$ |
|---|---|---|---|---|
| DIABETES | 250.* | 0.214 | 0.261 | 0.068 |
| CHRONIC KIDNEY DISEASE (CKD) | 585.1–9 | 0.106 | 0.119 | 0.015 |
| CHRONIC OBSTRUCTIVE PULMONARY DISEASE (COPD) | 496 | 0.069 | 0.136 | 0.017 |
| ASTHMA | 493.20–22 | 0.051 | 0.085 | 0.079 |
| HYPERTENSION (HTN-HEART) | 402.* | 0.001 | 0.028 | 0.006 |
| OSTEOARTHRITIS | 715.96 | 0.0197 | 0.061 | 0.034 |

### A.1   Dataset Details

**MIMIC Dataset**   The MIMIC III dataset includes a patient population of 46,520. There are 651,047 positive codes within 64,314 hospital admission records (HADM IDs). We have implemented an 80/20 split for training and testing purposes. Specifically, this allocates 12,862 records for testing and the remaining 51,451 for training. The histograms in Figure 6 indicate the density distribution of feature number per record. The dimension is $N = 1042$.

**Dataset $\mathcal{D}_1$**   The first dataset, denoted by $\mathcal{D}_1$, includes a patient population of size 1,670,347. We split the whole dataset into 100K for validation, 2000K for testing and the rest 1,370, 347 for training. The number of codes per patient is relatively small, as indicated by the histogram of feature number per record in Figure 8. We only consider medical codes with prevalence in corpus larger than 1.6e-5. The dimension is $N = 993$.

**Dataset $\mathcal{D}_2$**   The second dataset, denoted by $\mathcal{D}_2$, includes a patient population of size 1,859,536. We split the whole dataset into 100K for validation, 2000K for testing and the rest 1,559,536 for training. The number of codes per patient is relatively large, as indicated by the histogram of feature number per record in Figure 2. Although, dataset $\mathcal{D}_2$ has relatively denser feature, the prevalence of six chronic diseases we are interested in is pretty low. $\mathcal{D}_2$ contains diagnose codes (ICD), procedure codes (CPT) and medication codes (GEN). The number of ICD codes, CPT codes and GEN codes are 993, 690 and 1000 respectively. The total dimension is 2683.

### A.2   Model Architecture Detail

The denoise model in this paper has a uniform architecture, illustrated in Figure 4. The architecture we propose is tailored for tabular EHRs as it is non-sequential data. While the architecture proposed in multinomial diffusion Hoogeboom et al. (2021b) is designed for sequential data, where neighboring dimensions have semantic correlation. The tabular EHR datasets in our paper don't have such property. Therefore, we propose a novel transformer-based model for tabular EHRs. One bottleneck of transformer models is that the computational complexity of the attention module is quadratic to the dimension of input data. We adopt an efficient block based on Wang et al. (2020), whose attention operation has linear complexity with respect to the dimension of input data.

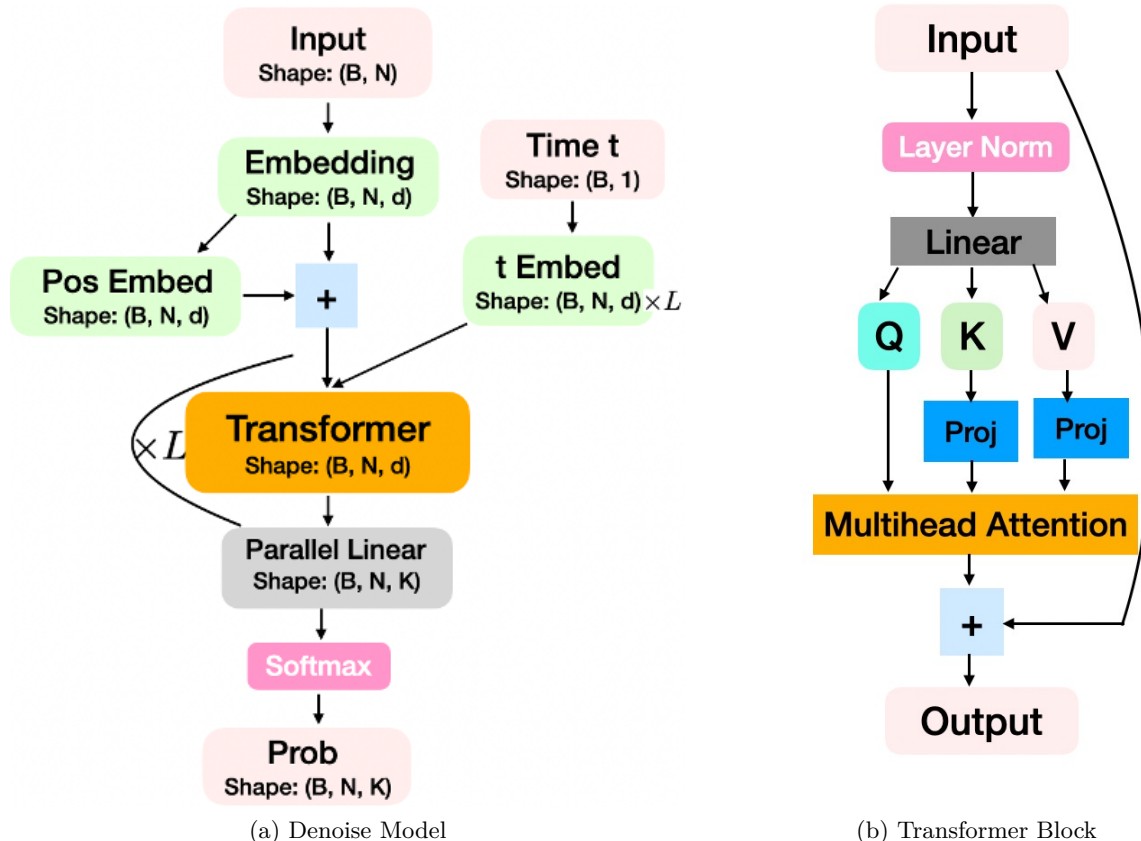

(a) Denoise Model                                    (b) Transformer Block

Figure 4: Architecture of our denoise model. (b) provides the detail of transformer block which has linear complexity with respect to the dimension of input. Axial positional embedding is employed to encode the positional information. We employ sinusoidal positional embedding to time $t$ to the time embedding and then use a two-layer MLPs to map the time embedding into hidden state. In the first layer of the two-layer MLP, we use Softplus activation function. We apply L times of such two-layer MLP to get the hidden state of time embedding to yield the input of each transformer block, as indicated in (a). Positional embedding is added to the embedding of discrete inputs. The input has dimension N and B means the batch size. For notation simplicity, we use all dimension of tabular data has K categories. We use one-hot representation and therefore, the output of the denoise model has shape (B, N, K). The shape of intermediate layers is provided in (a). In (b), "Proj" denotes the projection operation proposed in Linformer Wang et al. (2020), which induces the linear complexity of the attention module with respect to the input dimension $N$. The projection dimension is set as the default value 128 for all experiments in this paper.

### A.3    Hyper-parameters

**Hyper-parameters on MIMIC dataset**    Since the MIMIC III dataset is relatively small, we use a relative small model to train our EHR-D3PM to avoid overfitting to the training data. The hidden dimension 256. The number of multi-attention heads is 8. The number of transformer layers is 5. The number of diffusion steps is 500.

In the optimization phase, we adopt adamW optimizer, and the weight decay in adamW is 1.e-5. The learning rate is 1e-4 and batch size is 256. The beta for expentialLR in learning rate schedule is 0.99. The number of training epochs is 100. It takes less than three hours to finish training this model on A6000 with 48G memory.

**Hyper-parameters on datasets $\mathcal{D}_1$ and $\mathcal{D}_2$** The denoise model for datasets $\mathcal{D}_1$ and $\mathcal{D}_2$ are the same. As datasets $\mathcal{D}_1$ and $\mathcal{D}_2$ are large, we use a relatively large model. The number of multi-attention heads is 8. The hidden dimension is 512. The number of transformer layers $L$ is 8. The number of diffusion steps is 500.

The optimization parameters for both datasets $\mathcal{D}_1$ and $\mathcal{D}_2$ are also the same. In the optimization phrase, we adopt adamW optimizer. The learning rate is 1.0e-4 and batch size is 512. The weight decay in adamW is 1.0e-5. The beta for expentialLR in learning rate schdule is 0.99. The number of training epochs is 40. It takes one and half day to train one model on A100 with 80G memory.

**Hyper-parameters of baseline EHRDiff** To have a fair comparison with diffusion baseline EHRDiff, we use the same hyper parameters as our proposed diffusion model on all three datasets. The number of diffusion steps in EHRDiff is also 500 and the number of layers in EHRDiff is also 5. The other hyper parameters use the default values in the github implementation of EHRDiff.

**Hyper-parameters of baselines TabSyn and TabDDPM** To have a fair comparison with baselines TabSyn and TabDDPM, the number of diffusion steps in TabSyn and TabDDPM are also 500 and the number of layers in denoising models of TabSyn and TabDDPM is also 5. For other hyper parameters, we use the default parameters provided on the official implementation of TabSyn and TabDDPM. When training TabSyn on our datasets, we found that TabSyn works quite well in the first 100 dimensions of our datasets. But training TabSyn in the first 200 dimensions of our datasets, we found the performance significantly degenerates. The dimensions of seven datasets used in the paper of TabSyn are all less than 50.

### A.4 Evaluation Metrics

**MMD** The empirical MMD between two distributions P and Q is approximated by

$$\text{MMD}(P, Q) = \frac{1}{m} \sum_{\gamma=1}^{m} \hat{\text{MMD}}_{k_\gamma}(P, Q), \tag{6}$$

where $k_\gamma$ is a kernel function; m is number of kernels; $\hat{\text{MMD}}_{k_\gamma}(P, Q)$ is estimated by samples $\{\mathbf{x}_i\}_{i=1}^n \sim P$ and $\{\mathbf{x}_i'\}_{i=1}^n \sim Q$ as follows,

$$\hat{\text{MMD}}_{k_\gamma}(P, Q) = \frac{1}{n(n-1)} \left[ \sum_{i \neq j} k_\gamma(\mathbf{x}_i, \mathbf{x}_j) + \sum_{i \neq j} k_\gamma(\mathbf{x}_i', \mathbf{x}_j') \right] - \frac{1}{n^2} \sum_{i,j} k_\gamma(\mathbf{x}_i, \mathbf{x}_j').$$

In our evaluation, we use Gaussian RBF kernel $k_\gamma$,

$$k_\gamma(\mathbf{x}, \mathbf{x}') = \exp\left( - \frac{\|\mathbf{x} - \mathbf{x}'\|^2}{2h_\gamma^2} \right)$$

with bandwidth $h_\gamma = \text{Avg} * 2^{(\gamma - m/2)}$, where Avg is the average of pairwise L2 distance between all samples. We choose $m = 5$ and thus $\gamma \in \{1, 2, 3, 4, 5\}$.

### A.5 Hyper-parameters of Classifier Models on Downstream Tasks

For the downstream tasks, we used a light gradient boosting decision tree model (LGBM) (Ke et al., 2017) as it had uniformly robust prediction performance on all downstream tasks. In all experiments, we set the hyper-parameters of LGBM as follows: n_estimators = 1000, learning_rate = 0.05 max_depth = 10, reg_alpha = 0.5, reg_lambda = 0.5, scale_pos_weight = 1, min_data_in_bin = 128. We also experiment with various sets of hyper parameters which will induce the same conclusion we have in this paper.

## B Additional Experiments

Due to space limit, we leave a bunch of experiment results on appendix.

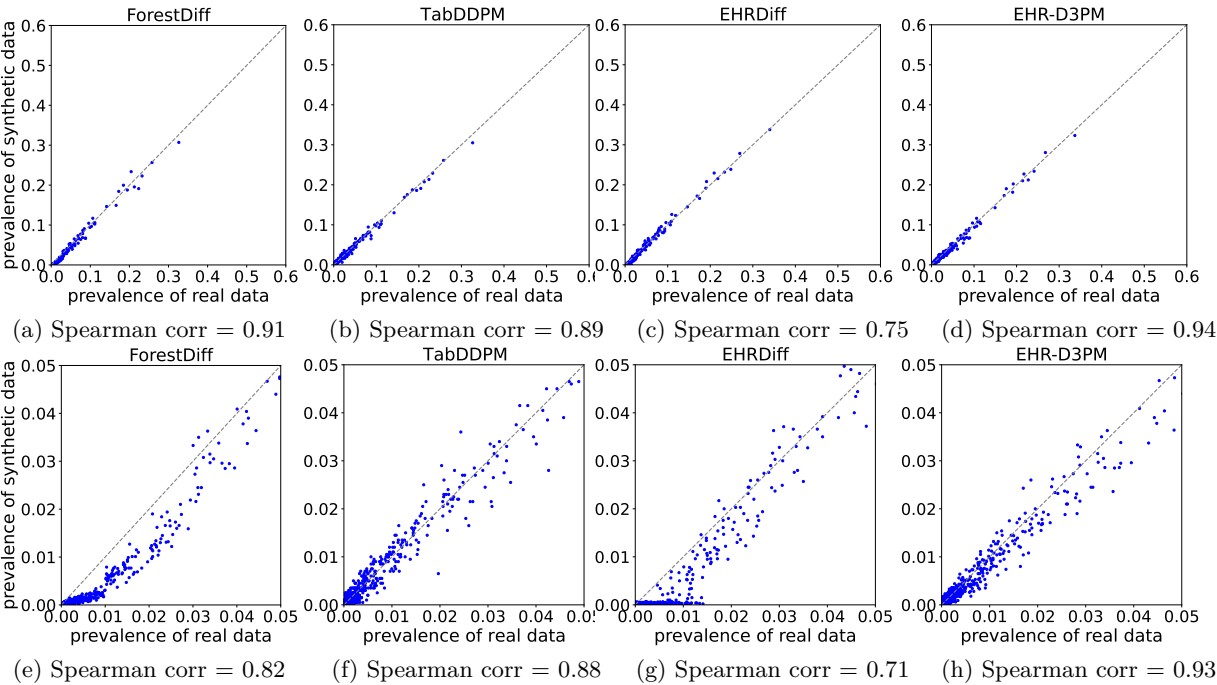

Figure 5: Comparison of prevalence in synthetic data and real data (MIMIC). The second row represents the prevalence of the first row in the low data regime. The prevalence is computed on 10K samples as the MIMIC dataset is relatively small. The dashed diagonal lines represent the perfect matching of code prevalence between synthetic data and real EHR data. Pearson correlations are very high for all methods and thus not used as a metric to compare different methods.

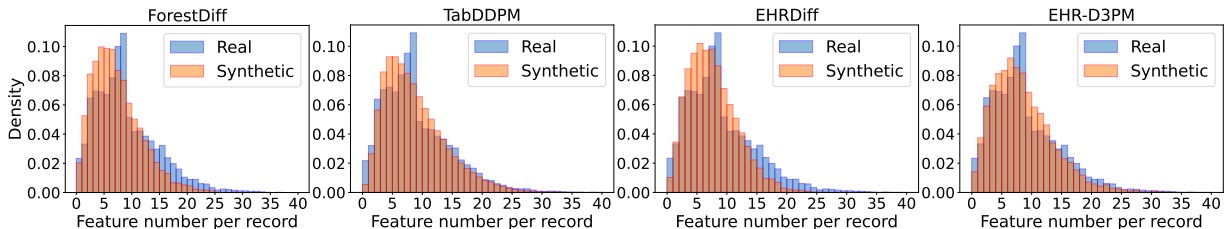

Figure 6: Density comparison of per-record feature number on synthetic and real data for the MIMIC dataset. The number of features per record is the sum of ICD codes present in each sample. The number of bins is 40, and the range of feature number values is (0, 40).

Table 6: Fidelity metrics (CMD, MMD and MCAD) on dataset MIMIC.

| METRICS | CMD($\downarrow$) | MMD($\downarrow$) | MCAD($\downarrow$) |
|---|---|---|---|
| MED-WGAN | $27.540_{\pm 0.628}$ | $0.078_{\pm 0.0089}$ | $0.1896_{\pm 0.0024}$ |
| EMR-WGAN | $26.658_{\pm 0.639}$ | $0.053_{\pm 0.0054}$ | $0.1546_{\pm 0.0017}$ |
| TABDDPM | $25.236_{\pm 0.684}$ | $0.010_{\pm 0.0016}$ | $0.1306_{\pm 0.0018}$ |
| EHRDIFF | $25.447_{\pm 0.485}$ | $0.009_{\pm 0.0013}$ | $0.1439_{\pm 0.0015}$ |
| FORESTDIFF | $24.173_{\pm 0.506}$ | $0.008_{\pm 0.0015}$ | $0.1375_{\pm 0.0016}$ |
| EHR-D3PM | $\mathbf{21.128}_{\pm 0.393}$ | $\mathbf{0.003}_{\pm 0.0004}$ | $\mathbf{0.1013}_{\pm 0.0010}$ |

### B.1 Additional experiments on unconditional generation

**Fidelity** Figure 5 and Figure 7 provide additional comparison of marginal distribution matching on dataset MIMIC and dataset $\mathcal{D}_1$. We have found that TabSyn works well in low dimension but has poor performance on our high dimensional datasets with sparse features. From the Spearman correlation in low prevalence regime, we can still see that our method significantly outperform baselines. Based on results in Figure 5, we observe that ForestDiff fails to provide an unbiased estimation of the distribution in the low prevalence regime ($<= 0.05$), which corresponds to rare conditions. From Figure 5, Figure 7 and Figure 1, we consistently observe the performance of EHRDiff significantly degenerates in the low prevalence regime. One reason we articulate is that the foundation of EHRDiff is designed for continuous distribution and cannot be readily applied to the generation of discrete data particularly on data of low prevalence regime. From Figure 6 and Figure 8, we can see that the histogram of feature number per record on synthetic data by our method provides the best matching to that of real data.

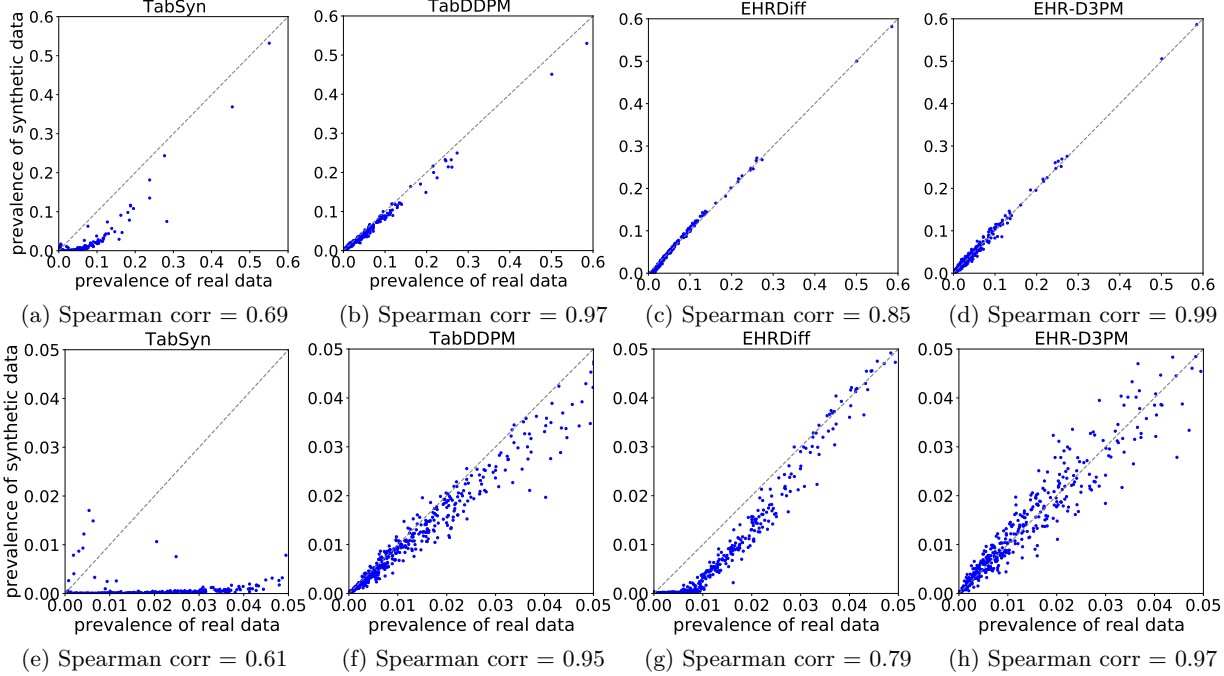

Figure 7: Comparison of prevalence in synthetic data and real data $\mathcal{D}_1$. The second row represents the prevalence of the first row in the low data regime. The prevalence is computed on 200K samples. The dashed diagonal lines represent the perfect matching of code prevalence between synthetic data and real EHR data. Spearman correlations between synthetic data and real data are reported.

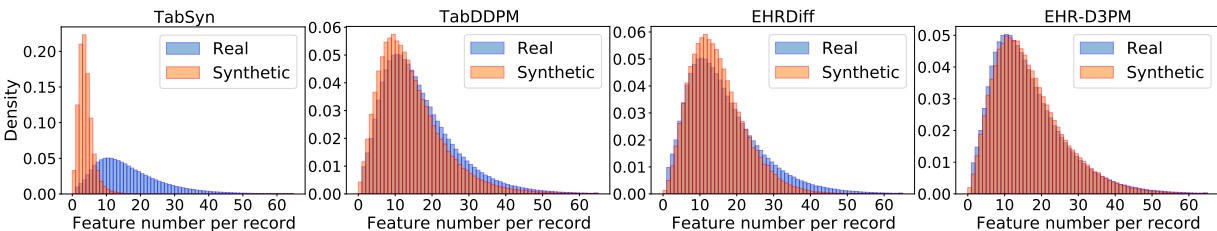

Figure 8: Density comparison of per-record feature number between synthetic data and real data $\mathcal{D}_1$. The number of features per record is computed summing the ICD codes present in each sample. The number of bins is 65 and the range of feature number values is (0, 65).

**Utility** We also apply our methods to downstream prediction tasks on dataset $\mathcal{D}_1$, where the prevalence of six chronic diseases is much lower. From Table 7 and Table 8, we can see that the accuracy of our prediction is still close to the prediction of classifier models trained on real data, which is the target classifier baseline. While other baselines have a much larger performance gap when compared with the ideal classifier. Particularly on rare diseases such as hypertension heart, the classifier trained on synthetic data by our model has 8% absolute improvement in AUPRC and AUROC over the strongest baseline on both $\mathcal{D}_1$ and $\mathcal{D}_2$. From the confidence intervals provided in Table 7, 8, 2 and 9, we confirm that such improvement over the baselines on both dataset $\mathcal{D}_1$ and dataset $\mathcal{D}_2$ are statistically significant.

## B.2 Additional experiments on guided generation

We provide additional experiment results on guided generation. We augment the real data with synthetic data generated by our sampling method and train a downstream classifier. We measure the performance of all classifiers on real test data. From Figure 9 and Figure 3, we see that the classifier trained on augmented training data with synthetic data, either by our unconditional sampling method or by our guided sampling method, consistently outperforms the classifier trained with original source data (vanilla baseline) and the conditional baseline EHRMGAN. In all cases, the relative increase of AUPRC over vanilla baseline is more than 3%; in the classification of COPD, the relative improvement over the vanilla baseline is more than 30%. This clearly indicates that our method can be applied to augment the training data of downstream classification tasks when the real dataset is scarce. More importantly, the data augmentation by guided sampler consistently outperforms the data augmentation with unconditional sampler. We observe this to be because the synthetic samples generated by guided sampling contain richer information in diseases of low prevalence. A most balanced training data with positive label will enhance the performance of classifiers and reduce the risk of over fitting to negative class.

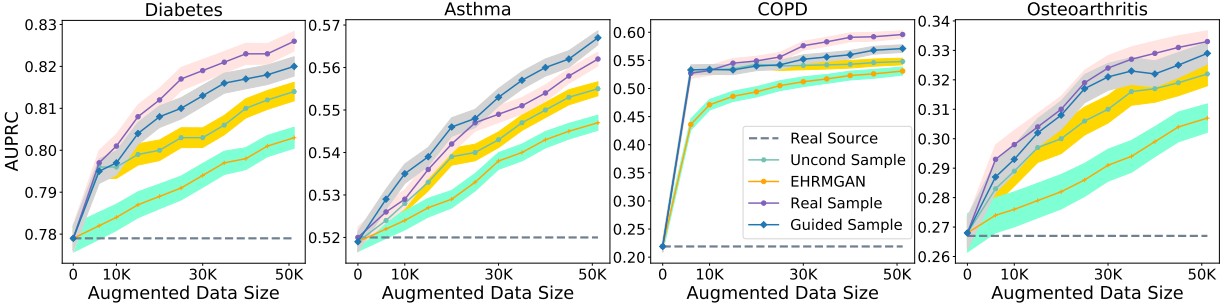

Figure 9: Synthetic data augmentation for disease classification from ICD codes based on dataset $\mathcal{D}_2$. The size of real source data for training LGBM classifier is 5000, as indicated in dashed line. We augment the original source training data with synthetic data to train LGBM classifier. "Uncond Samples" stands for the synthetic data generated by our unconditional sampler. Guided samples are synthetic data generated by our proposed guided sampler for each disease. To minimize noise from evaluation, we adopt 200K real test data to evaluate all experiments and report test AUROC for comparison. 80% of the test data are bootstrapped 50 times to compute 95% confidence intervals (CI), which is added as shaded region.

## C  Additional Details of `EHR-D3PM`

If $\mathbf{x}$ is a continuous variable, the most common way to sample from the posterior $p_{\boldsymbol{\theta}}(\mathbf{x}|\mathbf{c}) \propto p_{\boldsymbol{\theta}}(\mathbf{x}) \cdot p(\mathbf{c}|\mathbf{x})$ is using the following Langevin dynamics,

$$\mathbf{x}^{(k+1)} \leftarrow \mathbf{x}^{(k)} + \eta\tau_1 \nabla \log p(\mathbf{c}|\mathbf{x}) + \eta \nabla \log p_{\boldsymbol{\theta}}(\mathbf{x}) + \sqrt{\eta\tau_2}\boldsymbol{\epsilon}, \tag{7}$$

where $\boldsymbol{\epsilon} \sim \mathcal{N}(0, \mathbf{I})$. (7) has been applied in image (Dhariwal & Nichol, 2021) and recently be applied to EHR generation with Gaussian diffusion (He et al., 2023b). In practice, $\tau_2$ is always chosen to be zero in practice, and we will generally use $V(\mathbf{x}) := \log(p(c|\mathbf{x}))$ to replace the likelihood that we want to maximize in (7).

Table 7: Synthetic data utility. Disease prediction from ICD codes on real data $\mathcal{D}_1$. AUPRC is reported. We use synthetic data of 160K to train the classifier and 200K real test data to evaluate different methods. 80% of test data are bootstrapped for 50 times to compute for 95% confidence interval.

| | T2D | ASTHMA | COPD | CKD | HTN-HEART | OSTEOARTHRITIS |
|---|---|---|---|---|---|---|
| REAL DATA | $0.702_{\pm 0.002}$ | $0.288_{\pm 0.004}$ | $0.675_{\pm 0.002}$ | $0.806_{\pm 0.002}$ | $0.253_{\pm 0.003}$ | $0.296_{\pm 0.003}$ |
| MED-WGAN | $0.628_{\pm 0.002}$ | $0.149_{\pm 0.002}$ | $0.578_{\pm 0.002}$ | $0.722_{\pm 0.002}$ | $0.114_{\pm 0.001}$ | $0.192_{\pm 0.003}$ |
| EMR-WGAN | $0.656_{\pm 0.002}$ | $0.193_{\pm 0.002}$ | $0.603_{\pm 0.002}$ | $0.753_{\pm 0.002}$ | $0.151_{\pm 0.003}$ | $0.219_{\pm 0.003}$ |
| TABPPDM | $0.669_{\pm 0.003}$ | $0.238_{\pm 0.005}$ | $0.625_{\pm 0.004}$ | $0.778_{\pm 0.003}$ | $0.176_{\pm 0.006}$ | $0.238_{\pm 0.005}$ |
| EHRDIFF | $0.670_{\pm 0.002}$ | $0.232_{\pm 0.003}$ | $0.642_{\pm 0.002}$ | $0.782_{\pm 0.002}$ | $0.150_{\pm 0.002}$ | $0.245_{\pm 0.003}$ |
| EHR-D3PM | $\mathbf{0.693}_{\pm 0.002}$ | $\mathbf{0.263}_{\pm 0.003}$ | $\mathbf{0.655}_{\pm 0.002}$ | $\mathbf{0.796}_{\pm 0.002}$ | $\mathbf{0.229}_{\pm 0.003}$ | $\mathbf{0.278}_{\pm 0.003}$ |

Table 8: Synthetic data utility. Disease prediction from ICD codes on real dataset $\mathcal{D}_1$. AUROC is reported. We use synthetic data of 160K to train the classifier and 200K real test data to evaluate different methods. 80% of test data are bootstrapped for 50 times to compute 95% confidence interval.

| | T2D | ASTHMA | COPD | CKD | HTN-HEART | OSTEOARTHRITIS |
|---|---|---|---|---|---|---|
| REAL DATA | $0.808_{\pm 0.001}$ | $0.759_{\pm 0.002}$ | $0.867_{\pm 0.001}$ | $0.913_{\pm 0.001}$ | $0.832_{\pm 0.001}$ | $0.789_{\pm 0.001}$ |
| MED-WGAN | $0.757_{\pm 0.001}$ | $0.595_{\pm 0.002}$ | $0.806_{\pm 0.001}$ | $0.873_{\pm 0.001}$ | $0.625_{\pm 0.002}$ | $0.661_{\pm 0.002}$ |
| EMR-WGAN | $0.770_{\pm 0.001}$ | $0.642_{\pm 0.002}$ | $0.815_{\pm 0.001}$ | $0.885_{\pm 0.001}$ | $0.686_{\pm 0.002}$ | $0.689_{\pm 0.002}$ |
| TABPPDM | $0.787_{\pm 0.002}$ | $0.728_{\pm 0.004}$ | $0.851_{\pm 0.003}$ | $0.898_{\pm 0.002}$ | $0.746_{\pm 0.004}$ | $0.747_{\pm 0.005}$ |
| EHRDIFF | $0.789_{\pm 0.001}$ | $0.722_{\pm 0.002}$ | $0.856_{\pm 0.001}$ | $0.902_{\pm 0.001}$ | $0.714_{\pm 0.002}$ | $0.759_{\pm 0.002}$ |
| EHR-D3PM | $\mathbf{0.801}_{\pm 0.001}$ | $\mathbf{0.748}_{\pm 0.002}$ | $\mathbf{0.860}_{\pm 0.001}$ | $\mathbf{0.908}_{\pm 0.001}$ | $\mathbf{0.821}_{\pm 0.002}$ | $\mathbf{0.782}_{\pm 0.002}$ |

Table 9: Synthetic data utility. Disease prediction from ICD codes on real data $\mathcal{D}_2$. AUPRC is reported. We use synthetic data of 160K to train the classifier and 200K real test data to evaluate different methods. 80% of test data are bootstrapped 50 times to compute 95% confidence interval.

| | T2D | ASTHMA | COPD | CKD | HTN-HEART | OSTEOARTHRITIS |
|---|---|---|---|---|---|---|
| REAL DATA | $0.834_{\pm 0.002}$ | $0.581_{\pm 0.005}$ | $0.622_{\pm 0.009}$ | $0.733_{\pm 0.006}$ | $0.278_{\pm 0.011}$ | $0.373_{\pm 0.005}$ |
| MED-WGAN | $0.725_{\pm 0.003}$ | $0.496_{\pm 0.005}$ | $0.203_{\pm 0.007}$ | $0.166_{\pm 0.005}$ | $0.008_{\pm 0.001}$ | $0.223_{\pm 0.004}$ |
| EMR-WGAN | $0.734_{\pm 0.003}$ | $0.431_{\pm 0.004}$ | $0.402_{\pm 0.007}$ | $0.628_{\pm 0.009}$ | $0.134_{\pm 0.008}$ | $0.210_{\pm 0.004}$ |
| TABPPDM | $0.795_{\pm 0.004}$ | $0.558_{\pm 0.006}$ | $0.544_{\pm 0.008}$ | $0.692_{\pm 0.010}$ | $0.156_{\pm 0.012}$ | $0.327_{\pm 0.008}$ |
| EHRDIFF | $0.807_{\pm 0.003}$ | $0.549_{\pm 0.004}$ | $0.548_{\pm 0.007}$ | $0.690_{\pm 0.008}$ | $0.141_{\pm 0.009}$ | $0.319_{\pm 0.005}$ |
| EHR-D3PM | $\mathbf{0.821}_{\pm 0.002}$ | $\mathbf{0.572}_{\pm 0.004}$ | $\mathbf{0.607}_{\pm 0.007}$ | $\mathbf{0.714}_{\pm 0.007}$ | $\mathbf{0.226}_{\pm 0.008}$ | $\mathbf{0.348}_{\pm 0.006}$ |

Table 10: Fidelity metrics (CMD, MMD and MCAD) on dataset $\mathcal{D}_1$.

| METRICS | CMD($\downarrow$) | MMD($\downarrow$) | MCAD($\downarrow$) |
|---|---|---|---|
| MED-WGAN | $18.107_{\pm 0.125}$ | $0.075_{\pm 0.011}$ | $0.1871_{\pm 0.0016}$ |
| EMR-WGAN | $11.869_{\pm 0.108}$ | $0.018_{\pm 0.003}$ | $0.1625_{\pm 0.0013}$ |
| TABDDPM | $11.204_{\pm 0.164}$ | $0.021_{\pm 0.005}$ | $0.1542_{\pm 0.0019}$ |
| EHRDIFF | $23.208_{\pm 0.083}$ | $0.023_{\pm 0.003}$ | $0.1687_{\pm 0.0014}$ |
| EHR-D3PM | $\mathbf{7.692}_{\pm 0.026}$ | $\mathbf{0.012}_{\pm 0.001}$ | $\mathbf{0.0873}_{\pm 0.0007}$ |

For discrete data, (7) is intractable since we can't get the gradient backpropagation via $\nabla \log p_{\boldsymbol{\theta}}(\mathbf{x})$. In addition, (7) can't guarantee that $\mathbf{x}^{(k+1)}$ lies in the category $\{1, \ldots, K\}$ after update. Therefore, we need to

do Langevin updates on the latent space $\mathbf{z}_{L,t}$

$$\mathbf{y}^{(k+1)} \leftarrow \mathbf{y}^{(k)} - \eta \nabla_{\mathbf{y}^{(k)}} [\mathcal{D}_{\mathrm{KL}}(\mathbf{y}^{(k)}) - V_{\boldsymbol{\theta}}(\mathbf{y}^{(k)})] + \sqrt{2\eta\tau}\boldsymbol{\epsilon},$$

where $\mathbf{y}^{(k)}$ is the modification of $\mathbf{z}_{L,t}$, and $\mathcal{D}_{\mathrm{KL}}(\mathbf{y}^{(k)}) = \lambda \mathrm{KL}(p_\theta(\hat{\mathbf{x}}_0|\mathbf{y}^{(k)})||p_\theta(\hat{\mathbf{x}}_0|\mathbf{y}^{(0)}))$ is the KL divergence for regularization of the guided Markov transition. The gradient of the KL term plays a similar role as $\nabla p_{\boldsymbol{\theta}}(\mathbf{x})$ in (7). It will be interesting to leverage deterministic updates Liu & Wang (2016); Han & Liu (2018) to accelerate this process.

**Example:** Suppose that the context $c$ is to generate a EHR $\mathbf{x}$ such that $\mathbf{x}$ has diabete disease, i.e., the $k$-th token of $\mathbf{x}$ equals $[1,0]$, where $k = 156$ for MIMIC data set. Then we have $p(c|\hat{\mathbf{x}}_0) = 1$ if $k$-th token of $\mathbf{x}$ equals $[1,0]$ and $p(c|\hat{\mathbf{x}}_0) = 0$ otherwise. In addition $p_{\boldsymbol{\theta}}(\hat{\mathbf{x}}_0|\mathbf{y}^{(k)})$ is the $k$-th position output of softmax layer when input $\mathbf{y}^{(k)}$. Then we can compute the energy function as follows,

$$V_{\boldsymbol{\theta}}(\mathbf{y}^{(k)}) = \log(p(\mathbf{c}|\mathbf{y}^{(k)})) = \log \left( \sum_{\hat{\mathbf{x}}_0} p_{\boldsymbol{\theta}}(\hat{\mathbf{x}}_0|\mathbf{y}^{(k)}) p(\mathbf{c}|\hat{\mathbf{x}}_0) \right).$$

In all experiments of guided generation, the number of Langevin update steps is 10, $\eta = 0.1$ and $\lambda = 0.01$.

