# OpenReview forum: "Guided Discrete Diffusion for Electronic Health Record Generation"
_TMLR — Accepted by TMLR_

### Review · Reviewer_wcsW · 2024-12-20

**Summary Of Contributions:**

In the paper, a discrete diffusion model is described which allows for the generation of synthetic electronic health records. Both unconditional and conditional generation are supported. The model performs well on a number of benchmarks.

**Audience:**

Yes

**Claims And Evidence:**

Yes

**Requested Changes:**

See weaknesses and minor issues.


Minor issues:

Abstract, line 4: confidentially -> confidentiality

Page 2, first point in list: tailed -> tailored

Equation after (3):

-first line: dx_{1:T} should be removed

-last line: square brackets after expectation value missing

**Strengths And Weaknesses:**

Strengths:

*) The paper tackles and important question as to how useful ML models can be trained in the medical domain without endangering the anonymity of the patients.

*) The paper is well written.

*) The model is thoroughly evaluated on a range of benchmarks. It is especially interesting that models trained on data augmented with synthetic data perform better on downstream tasks.

Weaknesses:


*) The main concern I have about the paper lies in the novelty of the method. Does the difference of the presented model with respect to standard D3PMs lie in the specific way in which the data is represented, as outlined in Section 4.1? Or has the model architecture described in (4) also been newly developed for this model? I think it would be good to emphasize the differences more in the paper.

---

### Review · Reviewer_1i4h · 2025-01-04

**Summary Of Contributions:**

This paper applies D3PM for electronic health record generation, and compares the results with those by other generative methods.

**Audience:**

Yes

**Broader Impact Concerns:**

No concerns on the ethical implications

**Claims And Evidence:**

No

**Requested Changes:**

I think the authors should provide experimental results for electronic health record generation by recent discrete diffusion models and flow models as well as GPT type autoregressive models.

**Strengths And Weaknesses:**

Strengths: The authors conduct a lot of experiments.

Weaknesses: D3PM type methods were proposed three years ago, new methods like the work by Lou et al. (2024) proposed learning probability ratios, extending score matching (Song and Ermon, 2019) to discrete spaces, or discrete flows by Campbell et al. (2024) and Gat et al. (2024) have been proposed and provide better results than D3PMs. So why choose D3PM for electronic health record generation?

Andrew Campbell, Jason Yim, Regina Barzilay, Tom Rainforth, and Tommi Jaakkola. Generative flows on discrete state-spaces: Enabling multimodal flows with applications to protein co-design, ICML 2024.

Itai Gat, Tal Remez, Neta Shaul, Felix Kreuk, Ricky T. Q. Chen, Gabriel Synnaeve, Yossi Adi, Yaron Lipman, Discrete Flow Matching, NeurIPS 2024.

Aaron Lou, Chenlin Meng, and Stefano Ermon. Discrete diffusion language modeling by estimating the ratios of the data distribution. ICML 2024.

It is well known that GPT like autoregressive models has the best results for discrete data, but the authors didn't provide GPT type models' results as baseline.

The statement on the first paragraph of page 3 "The approach was further developed by Ho et al. (2020); Song et al. (2020), which incorporates categorical random variables using transition matrices with uniform probabilities." is wrong, since these two papers didn't address discrete data.

Many references are double cited such as:

Edward Choi, Siddharth Biswal, Bradley Malin, Jon Duke, Walter F Stewart, and Jimeng Sun. Generating
multi-label discrete patient records using generative adversarial networks. In Machine learning for healthcare
conference, pp. 286–305. PMLR, 2017a.

Edward Choi, Siddharth Biswal, Bradley Malin, Jon Duke, Walter F. Stewart, and Jimeng Sun. Generating
multi-label discrete patient records using generative adversarial networks. In Finale Doshi-Velez, Jim
Fackler, David Kale, Rajesh Ranganath, Byron Wallace, and Jenna Wiens (eds.), Proceedings of the 2nd
Machine Learning for Healthcare Conference, volume 68 of Proceedings of Machine Learning Research, pp.
286–305. PMLR, 18–19 Aug 2017b.

Dibaba Adeba Debal and Tilahun Melak Sitote. Chronic kidney disease prediction using machine learning
techniques. Journal of Big Data, 9(1):109, 2022a.

Dibaba Adeba Debal and Tilahun Melak Sitote. Chronic kidney disease prediction using machine learning
techniques. Journal of Big Data, 9(1):109, 2022b.

Chanjung Lee, Brian Jo, Hyunki Woo, Yoori Im, Rae Woong Park, and ChulHyoung Park. Chronic disease
prediction using the common data model: development study. JMIR AI, 1(1):e41030, 2022a.

Chanjung Lee, Brian Jo, Hyunki Woo, Yoori Im, Rae Woong Park, and ChulHyoung Park. Chronic disease
prediction using the common data model: development study. JMIR AI, 1(1):e41030, 2022b.

Many references are officially published at leading conferences but cited as arxiv papers.

There are grammatical errors, for example, "but their approach offers limited flexibility has privacy concerns."

---

### Review · Reviewer_pKPR · 2025-01-13

**Summary Of Contributions:**

The paper introduces a method for synthesizing electronic health records using a discrete diffusion model tailored for tabular medical code data. It introduces a flexible conditional generation mechanism that does not require training on data with context label. Extensive evaluation shows that the proposed approach outperforms various generative baselines.

**Audience:**

Yes

**Broader Impact Concerns:**

Nothing in particular.

**Claims And Evidence:**

Yes

**Requested Changes:**

1. Please discuss in main text how the proposed differs from EHRDiff (besides that it does not offer conditional generation), as this seems like a very related baseline.
2. I think it is necessary to address my 2nd weakness in the main text. Otherwise the proposed approach cannot be justified.

**Strengths And Weaknesses:**

Strengths:
1. The use of a discrete diffusion model for EHR data makes sense.
2. The proposed approach targets a practical problem.

Weaknesses:
1. Lack of novelty. Discrete diffusion and the classifier guidance technique in 4.3 are already proposed in previous works. The paper uses them to solve the EHR generation task. While this is a reasonable solution, it does not offer exciting new insights/approaches.
2. I am a bit skeptical about solving data scarcity problem (in EHR) by synthesizing. Learning an accurate generative model itself is data-hungry, as it requires accurate modeling of the data distribution. When there is a shortage of data, how to make sure the learned generative model approximates the (mostly) hidden data distribution well?

---

### Decision · Action_Editor_EgZm · 2025-04-03

**Recommendation:** Accept with minor revision

**Comment:**

The paper proposes a discrete diffusion model to generate synthetic electronic health records. Reviewers appreciated the important practical motivation on generating synthetic data that can be used for analysis and training ML models. The evaluation on multiple benchmarks is thorough.

However, reviewers noted that many newer methods have been proposed since D3PM that may provide better results. In addition to the feedback from the reviewers, I would request two changes:
1. As the authors write in the rebuttal, consider editing the title and abstract/intro to reflect that the primary goal is to solve a practical problem of generating tabular EHR. Avoid using terms such as "novel" method in the abstract and instead focus on the applied usecase.
2. Acknowledge and discuss the recent papers suggested by reviewers, so that future readers may be able to appreciate the pros/cons and assess those models for this task.

**Audience:**

Yes, results are practical and useful paper for applying ML in healthcare

**Claims And Evidence:**

* Method clearly described
* Extensive evaluation to support the claims about generation quality
* "Novelty" of the method--this claim is less supported

---

> ### Author Response · Authors · 2025-04-30
> **Thank you for your suggestions**
>
> We have made following changes accordingly:
> 1. emphasize that the primary goal is to solve a practical problem of generating tabular EHR; remove "novelty" phrase in our paper.
> 2. Add recent papers (suggested by  Reviewer 1i4h) with discussion of pros/cons in Section 2 (Related Work).
>
> Thank you!